# Molecular Switching through Chalcogen-Bond-Induced Isomerization of Binuclear (Diaminocarbene)Pd$^{II}$ Complexes [†]

Roman A. Popov, Alexander S. Novikov [ID], Vitalii V. Suslonov and Vadim P. Boyarskiy *[ID]

Institute of Chemistry, Saint Petersburg State University, 26 Universitetskii Prospect, Petergof, St. Petersburg 198504, Russia; pra.peterburg@yandex.ru (R.A.P.); ja2-88@mail.ru (A.S.N.); v.suslonov@spbu.ru (V.V.S.)
* Correspondence: v.boiarskii@spbu.ru
† In commemoration of the 300th anniversary of Saint Petersburg State University's founding.

**Abstract:** Binuclear diaminocarbene complexes, which form as a regioisomer mixture in the reaction between isocyanide–palladium(II) complex *cis*-[PdCl$_2$(CNXyl)$_2$] and 1,3-thiazol-2-amine, are able to exchange an anionic chloride ligand with other halides, such as Br or I. This process also affords binuclear complexes as mixtures of kinetically and thermodynamically controlled regioisomers. In CDCl$_3$ solutions, we observed interconversion of kinetically and thermodynamically controlled regioisomers. The results of the DFT calculations revealed that in CHCl$_3$ solution, each pair of the isomers exhibited two different types of chalcogen bonding such as S···X or S···N; the presence of CBs for two complexes in the solid state was also proven through X-ray crystallographic study. Based on the combined experimental and theoretical data, it could be concluded that thermodynamic favorability for the formation of thermodynamically controlled regioisomers increases in the Cl < Br ≈ I row and correlate well with the energy difference between S···N and S···X (X = Cl, Br, I) chalcogen bonds in kinetically and thermodynamically controlled products. This means that it is possible to change the structure of metallocycles in binuclear diaminocarbene complexes by simply replacing one halide ligand with another.

**Keywords:** chalcogen bonding; (Diaminocarbene)Pd$^{II}$ complexes; regioisomerization; halogen-dependent equilibrium



## 1. Introduction

Chalcogen bonding, or a chalcogen bond (CB), is one of the varieties of σ-hole non-covalent interactions [1,2]. This interaction consists of bonding the region of low electron density lying on the axis of a covalent bond between an electron withdrawing group (EWG) and an element outside of this bond (σ-hole), and the region of increased electron density [3]. In the case of the chalcogen bond, sulfur acts as this element. The region of increased electron density is most often due to the proximity to an element that has a lone electron pair. Such interactions are denoted σ-lp.

Recently, this phenomenon has been the subject of much research in the fields of polymer science [4–6], drug development [7,8], biochemistry [9–12], crystal engineering [13–17], and supramolecular chemistry [18–20]. The reason for this is that chalcogen bonds (as well as other non-covalent interactions) determine many practically significant physicochemical properties of chalcogen-containing compounds: conformational stabilization [21], molecular assembly [22–24], protein folding [25,26], molecular recognition [9,27], and even reactivity [28,29]. Of course, the heavy chalcogen atoms—namely selenium and tellurium– are better CB donors than sulfur due to their higher polarizability [30,31]. However, the much more common sulfur-containing compounds play a much more important role in applied research. This causes an ever-increasing interest in the study of CBs, including S-centers.

The best known types of CB involving S atoms are S···N, S···O, and S···S interactions, which are often found both in biomolecules [10,32] and in organic [33–35] and coordination compounds [36–38]. Most of the examples described relate to the stabilization by CB of a certain conformation in intermediates or in starting compounds. These are papers on stabilization of the Pummerer reaction intermediate by inter- and intramolecular S···O contacts [39]; the use of chiral derivatives of isothiourea as organocatalysts, forming intramolecular S···O CB in reaction intermediates [40–42]; photochromic cyclization of 2,3-dithiazolylbenzothiophene [43]; assistance in intramolecular redox cyclization of S-containing diazene [44]; and bond-switching rearrangement of substituted isothiazoles and isothiadiazoles [45] to form, in both cases, intramolecular S···N-CB in the starting compounds. Other reports reveal the effect of CB on the reaction center, namely, hydrodeboronation of 5-thiazolylboronic acid [46], facilitation of S–S bond cleavage in benzo-1,2-dithiolan-3-one-1-oxides by formation of S···O contacts [47], and the Mann–Pope reaction [48], which are due to intramolecular S···O interactions. So far, CBs of sulfur atoms with halogens have been less studied, although the interest in such interactions is undeniable. For example, in the previously mentioned work on benzo-1,2-dithiolan-3-on-1-oxides, S–S bond cleavage can also be facilitated by the interaction of S···Cl.

We have previously shown [36] that the cause of a chemical reaction can be not only CB itself, but also the difference between two CBs. Binuclear diaminocarbene complexes formed upon treatment of *cis*-[PdCl$_2$(CNXyl)$_2$] (Xyl = 2,6-Me$_2$C$_6$H$_3$) with 1,3-thiazol-2-amines exist as a mixture of regioisomers (Scheme 1), one of which is the product of the kinetic control (isomer **I**), and the other is the thermodynamic one (isomer **II**). In CHCl$_3$ solutions, isomer **I** is reversibly converted to isomer **II**. According to theoretical calculations, the reason for this transformation is that chalcogen bond S···N in **II** is 1.8–2.3 kcal/mol more stable than chalcogen bond S···Cl in **I**.

**Scheme 1.** The reactions of *cis*-[PdCl$_2$(XylNC)$_2$] with thiazol-2-amines.

In this work, we have shown that the equilibrium in this system can be further shifted by a change of the halide ligand at the Pd$^{II}$-center. The latter can be caused by a change of the energy difference between two types of S···Hal and S···N intramolecular chalcogen bonds.

## 2. Results and Discussions

### 2.1. Synthesis of Binuclear Diaminocarbene Complexes

As has been previously reported [36], the reaction of *cis*-[PdCl$_2$(CNXyl)$_2$] (**1**) with 1,3- thiazol-2-amine (**2**) in a molar ratio 1:1.5 in CH$_2$Cl$_2$ at RT affords a mixture of two binuclear aminocarbene palladium(II) complexes bearing chloride anion ligands (Scheme 1). These products are the kinetically controlled regioisomer **3a** and the thermodynamically controlled regioisomer **3b**.

The treatment of the reaction mixture containing the chloride complexes **3a** and **3b** with two-fold excess of AgOTf in CH$_2$Cl$_2$ at RT, followed by the addition of KBr or KI, affords a mixture of two new binuclear aminocarbene products for each case, corresponding to kinetically (**4a** or **5a**) and thermodynamically (**4b** or **5b**) controlled regioisomers (Scheme 2). The mixture is enriched with the thermodynamically controlled regioisomer (Scheme 3).

**Scheme 2.** Obtaining aminocarbene complexes with different halogens.

**Scheme 3.** Regioisomerization of the obtained aminocarbene complexes.

## 2.2. The Regioisomerization

As the reaction proceeds, a mixture of both isomers (**a** and **b**) is formed. However, over time, the mixture is enriched with the thermodynamically controlled regioisomer (Scheme 3).

The isomerization of the kinetically to the thermodynamically controlled regioisomers was monitored via $^1$H NMR in CDCl$_3$ (Table 1). The isomerization of **3a** was slow. At 45 °C, the equilibrium was reached after 10 days (Figure 1). In contrast to the chlorine-containing complexes, the isomerization of **4a** (Figure 2) and **5a** (Figure 3) proceeded much faster; at 45 °C, it took 30 h.

**Table 1.** Ratios and constants based on $^1$H NMR monitoring at 45 °C.

| Mixture | Equilibrium Ratio | Equilibrium Constant | $-\Delta G_{(CHCl3)}$, kcal/mol |
|---|---|---|---|
| **3a/3b** [36] | 13/87 | 6.7 | 1.2 |
| **4a/4b** | 4/96 | 24 | 2.0 |
| **5a/5b** | 4/96 | 24 | 2.0 |

The observed regioisomerization can proceed only with cleavage of the Pd−N and C−N bonds of the aminocarbene fragment. The kinetical study of the regioisomerization of the similar binuclear diaminocarbene complexes showed that it is a first-order reaction; that is, it occurs intramolecularly [37]. It was concluded that the rate-determining step of the isomerization is breaking the carbon–nitrogen bond in the carbene fragment of the binuclear complex.

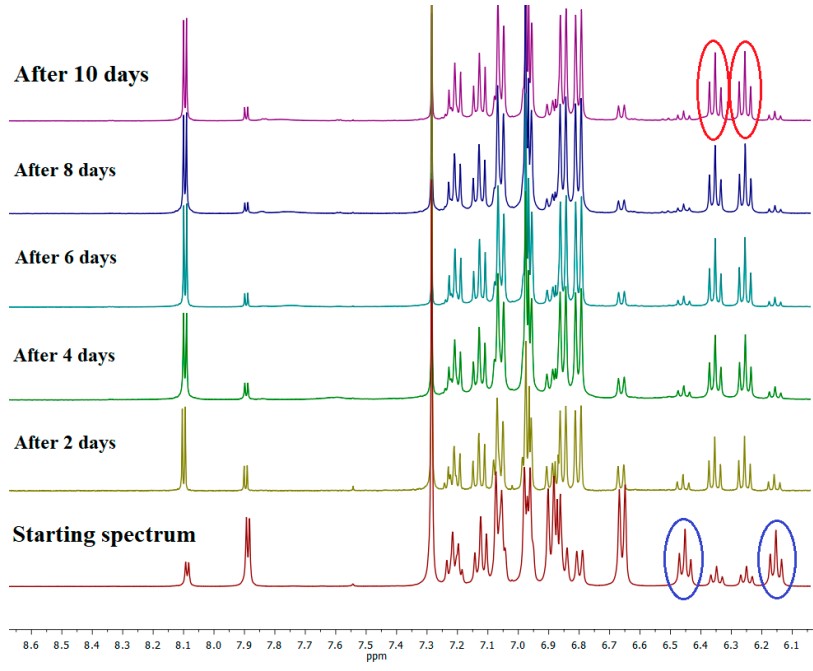

**Figure 1.** Monitoring of **3a**→**3b** isomerization at 45 °C (CDCl$_3$, $^1$H NMR data). Blue circles correspond to signals of Xyl-C and Xyl-B aromatic protons (H$^4$) for complex **3a**. Red circles correspond to signals of Xyl-C and Xyl-B aromatic protons (H$^4$) for complex **3b**.

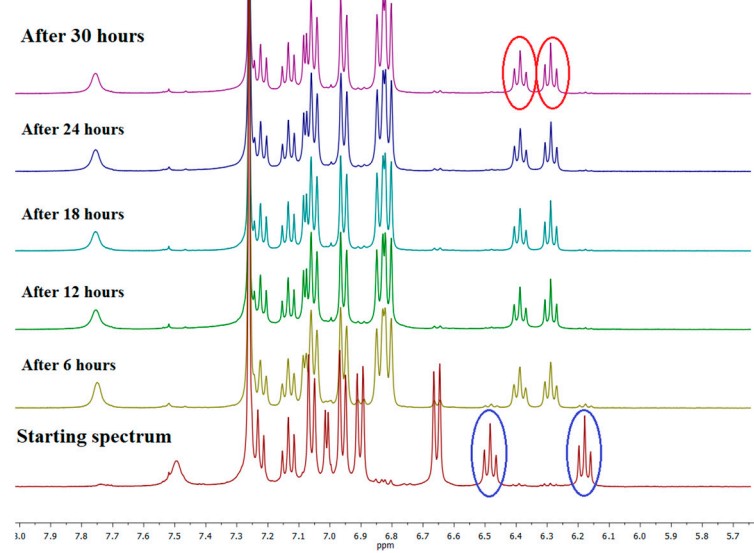

**Figure 2.** Monitoring of **4a**→**4b** isomerization at 45 °C (CDCl$_3$, $^1$H NMR data). Blue circles correspond to signals of Xyl-C and Xyl-B aromatic protons (H$^4$) for complex **4a**. Red circles correspond to signals of Xyl-C and Xyl-B aromatic protons (H$^4$) for complex **4b**.

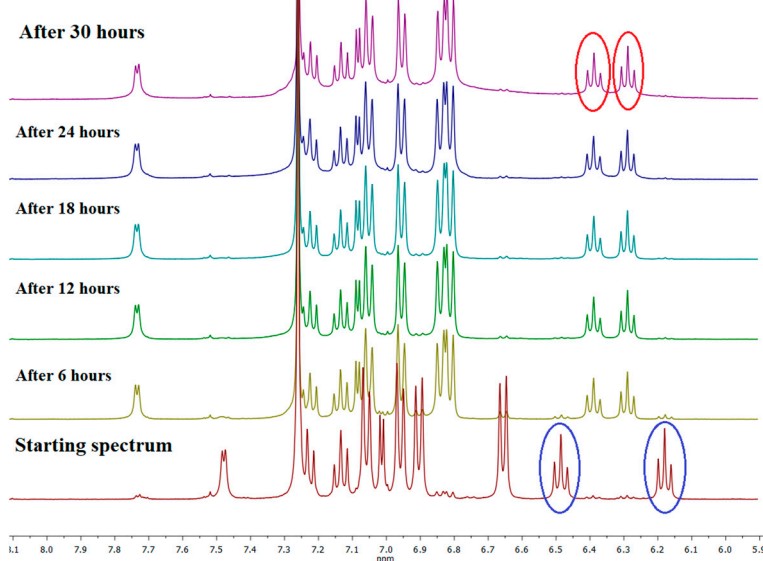

**Figure 3.** Monitoring of **5a**→**5b** isomerization at 45 °C (CDCl$_3$, $^1$H NMR data). Blue circles correspond to signals of Xyl-C and Xyl-B aromatic protons (H$^4$) for complex **5a**. Red circles correspond to signals of Xyl-C and Xyl-B aromatic protons (H$^4$) for complex **5b**.

Data on the isomeric ratios and the equilibrium constants are presented in Table 1. It can be seen from the obtained data that the bromide and iodide complexes are relatively quickly converted into the almost pure regioisomers, **b**. This means that it is impossible to isolate regioisomers, **a**, from the initially formed mixtures. To obtain the pure complexes **4a** and **5a**, we applied the following method. The replacement of the halogen was carried out not for mixture **3a/b**, but for the pure isomer **3a** (Scheme 4). Since the reaction was carried out at RT for a short time (10 min), regiomerization did not have time to occur, and we were able to isolate the pure complexes **4a** and **5a**.

**Scheme 4.** Obtaining pure kinetically controlled regioisomers **4a** and **5a**.

Such a dependence of the structure of binuclear complexes on the nature of the halogen makes it possible to obtain the desired specific regioisomer; that is, by changing the halide ligand, we can switch the mutual arrangement of atoms in the bicyclic structure.

### 2.3. Characterization of the Regioisomers **4a,b**–**5a,b**

Complexes **4**–**5a,b** were obtained as yellow solids and characterized by elemental (C, H, N) analysis (as the regioisomer mixtures), high-resolution ESI$^+$–MS (as the regioisomer mixtures), FTIR, and 1D ($^1$H, $^{13}$C{$^1$H}) and 2D ($^1$H,$^1$H-COSY, $^1$H,$^1$H-NOESY, $^1$H,$^{13}$C-HSQC, $^1$H,$^{13}$C-HMBC) NMR spectroscopies. The assignments of the $^1$H and $^{13}$C signals were performed with $^1$H,$^1$H-COSY, $^1$H,$^1$HNOESY, $^1$H,$^{13}$C-HSQC, and $^1$H,$^{13}$C-HMBC NMR methods. For all isomer mixtures, the HRESI$^+$ mass spectra displayed a fragmentation pattern corresponding to [M − Hal]$^+$ with the characteristic isotopic distribution.

The FTIR spectra for all complexes exhibited one strong and broad ν(C≡N) band in the range 2210–2190 cm$^{-1}$ from isocyanide ligands.

The $^1$H NMR spectra of the complexes displayed a set of overlapping and individual signals in the δ range 6.10–7.25 ppm corresponding to the twelve aromatic C–H protons of the xylyl groups and signals of the thiaazaheterocyclic fragment in the δ range 8.25–6.80 ppm. Signals of the Me groups were observed in the δ range 2.45–2.00 ppm.

The $^{13}$C NMR spectra of the complexes exhibited two signals of carbons in the NCN fragments in the δ range 150–195 ppm. These resonances belong to the typical range specific to Pd–C$_{carbene}$ ($\delta_C$ = 160–200 ppm) in acyclic diaminocarbenes [36,49,50].

In addition, the XRD data for complexes **4b** and **5b** were obtained. The plots of the structures **4b** and **5b** with the highlighted intramolecular chalcogen bonds are presented in Figure 4. Selected bond lengths and angles are given in Table 2. The metal centers have a slightly distorted square planar structure, and the halogen and nitrogen atoms bonded to the metal center are in the *cis* position to each other. In the NCN fragments bonded to the metal atom, the length of one of the CN bonds is in the range 1.407–1.444 Å; that is, it is close in length to the simple C–N bond (1.451(1) Å) [51]. The other has a length of 1.259–1.290 Å, which corresponds to the length of the double C=N bond (1.260(1) Å) [51]. All bond lengths and angles are similar to those for the previously reported thiazole-2-amine-based analogues [36].

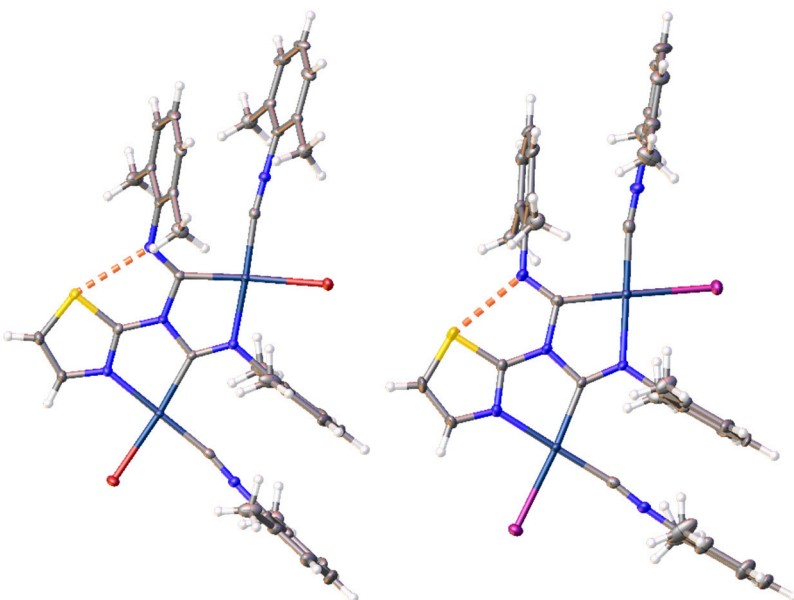

**Figure 4.** Views of XRD structures of **4b** (**left**) and **5b** (**right**). Thermal ellipsoids are drawn at the 50% probability level. Yellow stripped lines indicate S⋯N intramolecular chalcogen bonds.

**Table 2.** Selected bond lengths (Å) and angles (°) for compounds **4b**, **5b**.

| Complex | 4b | 5b |
|---|---|---|
| d(Pd1–C$_{carbene}$) | 2.014(2) | 2.034(3) |
| d(N2–C$_{carbene}$) | 1.407(3) | 1.409(4) |
| d(N3–C$_{carbene}$) | 1.286(3) | 1.290(4) |
| d(Pd2–C$_{carbene2}$) | 1.986(2) | 2.007(3) |
| d(N2–C$_{carbene2}$) | 1.438(3) | 1.444(4) |
| d(N4–C$_{carbene2}$) | 1.265(3) | 1.259(4) |
| ∠(N1–Pd1–Br1) | 94.94(6) | |
| ∠(N3–Pd2–Br2) | 97.05(6) | |
| ∠(N1–Pd1–I1) | | 95.95(8) |
| ∠(N3–Pd2–I2) | | 97.73(7) |

Checking of the XRD data indicates the existence of chalcogen bonding S···N in the bromide and iodide complexes **4b** and **5b** in solid state. Indeed, the distances between the S and N atoms in **4b** and **5b** (2.708(2) Å and 2.704(3), respectively) are shorter than their van der Waals radii sum (viz., 3.35 Å according to Bondi's definition) [52] and very close to the same contact in the previously reported structure of the thiazole-2-amin-based binuclear aminocarbene chloride complex **3b** (2.672 Å).

### 2.4. Analysis of the Intramolecular Chalcogen Bonding

We performed quantum chemical calculations and carried out topological analysis of the electron density distribution (aka AIM analysis) [53] to confirm the existence of these non-covalent interactions and quantitatively estimate their energies. Note that we already used such an approach in studies of these non-covalent contacts for the previously reported thiazole-2-amin-based binuclear aminocarbene chloride complexes **3a** and **3b** [36]. The results of the current quantum chemical study are given in Table 3, and the Laplacian of electron density distribution $\nabla^2\rho(\mathbf{r})$ contour line diagram, together with surfaces of zero-flux and bond paths for **4b**, are illustrated in Figure 5. The results of the AIM analysis reveal that the S···N chalcogen bonds in **4b** and **5b** are slightly weaker than in the chloride complex **3b** [36] in the solid state (4.9–5.3 kcal/mol vs. 5.1−6.0 kcal/mol).

**Table 3.** Electron density [$\rho(\mathbf{r})$], Laplacian of electron density [$\nabla^2\rho(\mathbf{r})$], energy density [$H_b$], potential energy density [$V(\mathbf{r})$], and Lagrangian kinetic energy [$G(\mathbf{r})$] (in Hartree) at the bond-critical points (3, −1), associated with chalcogen bonding S···X (X = Cl, Br, I) and S···N in **3–5a** and **3–5b** in solid state and chloroform solution, contact length [*l*] (in angstroms), and estimated energies for these contacts [$E_{int}$] (in kcal/mol).

| Compound | Contact | $\rho(\mathbf{r})$ | $\nabla^2\rho(\mathbf{r})$ | $H_b$ | $V(\mathbf{r})$ | $G(\mathbf{r})$ | $E_{int}$ [a] | $E_{int}$ [b] | *l* |
|---|---|---|---|---|---|---|---|---|---|
| | | | | Solid-state structures | | | | | |
| **3b** [c] | S···N | 0.026 | 0.080 | 0.001 | −0.019 | 0.019 | 6.0 | 5.1 | 2.67 |
| **4b** | S···N | 0.024 | 0.076 | 0.001 | −0.017 | 0.018 | 5.3 | 4.9 | 2.71 |
| **5b** | S···N | 0.024 | 0.076 | 0.001 | −0.017 | 0.018 | 5.3 | 4.9 | 2.70 |
| | | | | Optimized structures (CHCl$_3$) | | | | | |
| **3a** [c] | S···Cl | 0.015 | 0.051 | 0.002 | −0.009 | 0.011 | 3.0 | 2.8 | 3.14 |
| **3b** [c] | S···N | 0.024 | 0.071 | 0.001 | −0.016 | 0.017 | 5.0 | 4.6 | 2.73 |
| **4a** | S···Br | 0.016 | 0.049 | 0.002 | −0.009 | 0.011 | 2.8 | 3.0 | 3.22 |
| **4b** | S···N | 0.024 | 0.073 | 0.001 | −0.017 | 0.018 | 5.3 | 4.9 | 2.72 |
| **5a** | S···I | 0.011 | 0.031 | 0.001 | −0.005 | 0.006 | 1.6 | 1.6 | 3.58 |
| **5b** | S···N | 0.025 | 0.074 | 0.001 | −0.017 | 0.018 | 5.3 | 4.9 | 2.71 |

[a] $E_{int} = -V(r)/2$ [54]; [b] $E_{int}$ 0.429G(r) [55]; [c] Data taken from Ref. [36].

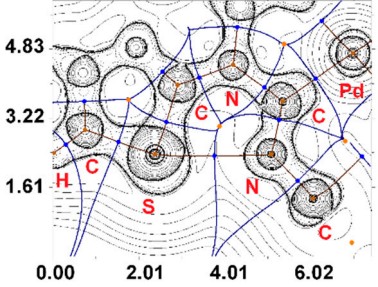

**Figure 5.** Laplacian of electron density distribution $\nabla^2\rho(\mathbf{r})$ contour line diagram together with surfaces of zero-flux and bond paths for **4b**. Nuclear critical points (3, −3) are indicated as pale brown dots, bond critical points (3, −1) are indicated as blue dots, ring critical points (3, +1) are indicated as orange dots, and length units are given in angstroms.

To estimate the effect of these non-covalent interactions on the chemical equilibrium of the **4–5a,b** complexes in the chloroform solution, we carried out geometry optimization of these two isomeric pairs using DFT calculations followed by additional AIM analysis, as well as theoretical analysis of relative stability for these isomer pairs in the chloroform solution. In all cases, AIM analysis indicated the presence of chalcogen bonds S···X (X = Br, I) for **4–5a** and S···N for **4–5b** in the optimized equilibrium model structures. Analysis of energy differences between the S···X (X = Cl, Br, I) and S···N intramolecular chalcogen bonds in the complex pairs **4–5a/4–5b** indicated that in all cases the S···N interaction was stronger than for S···X (X = Cl, Br, I) (Table 3). Note that elongation of S···X contacts in the optimized equilibrium model structures in the row Cl<Br<I correlates well with increasing of atomic radii in the same order (viz. Cl<Br<I) (Table 3). It was experimentally observed that the change of chloride ligand on bromide and iodide led to a shift of equilibrium towards complexes **3–5b** during the isomerization of complexes **3–5a/3–5b**, and this observation can also be supported by theoretical analysis of the relative stability of these isomer pairs in the chloroform solution; in accordance with the conducted quantum chemical calculations, the relative stability of **b** complexes with an S···N chalcogen bond also increased in the row Cl < Br ≈ I (Table 4).

**Table 4.** Values of experimental and theoretically calculated $-\Delta G_{318}$ for isomerization of complexes **4–5**.

| System | $-\Delta G_{exp}$ | $-\Delta G_{calcd}$ |
|:---:|:---:|:---:|
| **3a –> 3b** * | 1.2 | 3.2 |
| **4a –> 4b** | 2.0 | 5.4 |
| **5a –> 5b** | 2.0 | 5.8 |

* Data taken from Ref. [36].

## 3. Materials and Methods

### 3.1. General

Solvents, PdCl$_2$, AgOTf, heterocycle **2**, and xylyl isocyanide were obtained from commercial sources and used as received. Complex *cis*-[PdCl$_2$(CNXyl)$_2$] was synthesized via the procedure outlined in the literature [56]. Elemental analyses were carried out on a Euro EA 3028 HT CHNSO analyzer. ESI mass-spectra were recorded on a Bruker micrOTOF (Bruker, Billerica, MA, USA) spectrometer with CH$_2$Cl$_2$/MeOH mixture as a solvent. The analyses were carried out in positive ion mode (*m/z* range of 50–3000). The most intensive peak in the isotopic pattern was reported. IR spectra were recorded on Shimadzu FTIR 8400S (Shimadzu, Kyoto, Japan) instrument in KBr pellets (4000–400 cm$^{-1}$, resolution 2 cm$^{-1}$). The 1D NMR spectra were recorded on a Bruker Avance 400 (Bruker Corporation, Billerica, MA, USA) spectrometer; 2D NMR correlation experiments were recorded on a Bruker Avance II+ 500 MHz (UltraShield Magnet) (Bruker Corporation, Billerica, MA, USA) spectrometer. All NMR spectra were recorded in CDCl$_3$. X-ray diffraction data were collected at a single crystal diffractometer (Agilent Technologies, Santa Clara, CA, USA) (Oxford Diffraction) "Xcalibur" using Mo-K$\alpha$ ($\lambda$ = 0.71073 nm) radiation.

### 3.2. Synthesis of the Complexes

Synthesis of **3a**. The solid isocyanide complex **1** (0.2 mmol) and heterocycle **2** (0.3 mmol) were placed in a 10 mL flat-bottom flask and filled with 5 mL of CH$_2$Cl$_2$. The resulting suspension was stirred at room temperature for 2 days. As the reaction proceeded, the reaction mixture turned yellow and aminothiazole hydrochloride precipitate formed. The solution was filtered from the precipitate and dried under reduced pressure. The resulting mixture contained complexes **3a** and **3b**. The summary yield was 82%. For the further synthesis of complexes **4b** and **5b**, the obtained mixture of complexes **3a** and **3b** was used further without separation. The pure complex **3a** was synthesized as per the literature [36].

Synthesis of **4a** and **5a**. Pure complex **3a** (0.03 mmol) and silver triflate (0.06 mmol) were added to a 10 mL flat-bottom flask and filled with 2 mL of CH$_2$Cl$_2$. The resulting

suspension was put into a room temperature ultrasonic bath for 10 min. As the reaction proceeded, a yellow solution of complex **6a** and white precipitate of AgCl formed. The solution was filtered from the precipitate and then 1 mmol KBr (in the case of **4a**) or KI (in the case of **5a**) was added. The resulting suspension was stirred at room temperature for 10 min. After that, the solution was filtered from the precipitate and dried under reduced pressure.

Synthesis of **4a/b** (mixture) and **5a/b** (mixture). A mixture of complexes **3a** and **3b** (0.03 mmol) and silver triflate (0.06 mmol) was added to a 10 mL flat-bottom flask and filled with 2 mL of CHCl$_3$. The resulting suspension was put into a room temperature ultrasonic bath for 10 min. As the reaction proceeded, a yellow solution of mixture of complexes **6a** and **6b** and white precipitate of AgCl formed. The solution was filtered from the precipitate and then 1 mmol KBr (in the case of **4a,b**) or KI (in the case of **5a,b**) was added. The resulting suspension was stirred at room temperature for 10 min. After that, the solution was filtered from the precipitate and dried under reduced pressure.

Synthesis of **4b** and **5b**. The resulting mixture from the previous stage was dissolved in 2 mL of CHCl$_3$ and heated for 2 d at 45 °C. After that, the solution was dried under reduced pressure.

### 3.3. Characterisation of the Complexes

**3a/b mixtures**. Anal. calcd for C$_{39}$H$_{38}$Cl$_2$N$_6$Pd$_2$S·0.2CDCl$_3$: C, 50.59; H, 4.16; N, 9.03, found: C, 50.19; H, 4.12; N, 9.43. HRESI$^+$ MS: calcd for C$_{39}$H$_{38}$ClN$_6$Pd$_2$S$^+$ 871.0641, found $m/z$ 871.0611 [M − Cl]$^+$. IR (KBr, selected bands, cm$^{-1}$): $\nu$(C–H) 2920 (w), 2850 (w), $\nu$(C≡N) 2193 (s), $\nu$(C=N) 1634 (s), 1554 (s), $\delta$(C–H) 772 (m).

**3a**. Yield 68%. NMR spectra fully corresponding with the literature data [36]. $^1$H NMR ($\delta$, ppm, J/Hz): 2.04 (s, 6H, CH$_3$, Xyl-C), 2.21 (s, 6H, CH$_3$, Xyl-D), 2.26 (s, 6H, CH$_3$, Xyl-A), 2.42 (s, 6H, CH$_3$, Xyl-B), 6.13 (t, 1H, H$^4$, Xyl-C, J = 7.6), 6.43 (t, 1H, H$^4$, Xyl-B, J = 7.6), 6.63 (d, 2H, H$^{3,5}$, Xyl-C, J = 7.6), 6.82–6.88 (m, 2H, H$^{3,5}$, Xyl-B and 1H, H$^5$, thiazole), 6.93–6.96 (m, 2H, H$^{3,5}$, Xyl-D), 7.02–7.05 (m, 2H, H$^{3,5}$, Xyl-A), 7.10 (t, 1H, H$^4$, Xyl-D, J = 7.6), 7.16–7.21 (m, 1H, H$^4$, Xyl-A), 7.86 (d, 1H, H$^4$, thiazole, J = 4.0). $^{13}$C{$^1$H} NMR ($\delta$, ppm): 18.40 (2C, CH$_3$, Xyl-D), 18.46 (2C, CH$_3$, Xyl-A), 19.42 (2C, CH$_3$, Xyl-B), 19.84 (2C, CH$_3$, Xyl-C), 112.70 (1C, C$^5$, thiazole), 123.71 (1C, C$^4$, Xyl-C), 126.43 (1C, C$^1$, Xyl-A), 126.80 (2C, C$^{2,6}$, Xyl-C), 127.34 (2C, C$^{3,5}$, Xyl-D), 127.54 (2C, C$^{3,5}$, Xyl-C), 127.72 (2C, C$^{3,5}$, Xyl-A), 127.74 (2C, C$^{3,5}$, Xyl-B), 128.95 (1C, C$^4$, Xyl-B), 129.48 (1C, C$^4$, Xyl-D), 129.60 (1C, C$^4$, Xyl-A), 134.15 (4C, C$^{2,6}$, Xyl-A and Xyl-D), 134.13 (1C, C$^4$, thiazole), 136.29 (2C, C$^{2,6}$, Xyl-B), 142.51 (1C, C$^1$, Xyl-B), 149.71 (1C, C$^1$, Xyl-C), 164.16 (1C, C$^2$, carbene), 180.26 (1C, C$^2$, thiazole), 194.67 (1C, C$^1$, carbene). The signals of the isocyanide quaternary atoms and C$^1$ Xyl-D were not found even at high acquisition.

**3b**. NMR spectra fully corresponding with the literature data [36]. $^1$H NMR ($\delta$, ppm, J/Hz): 2.21 (s, 6H, CH$_3$, Xyl-D), 2.27 (s, 6H, CH$_3$, Xyl-C), 2.28 (s, 6H, CH$_3$, Xyl-A), 2.42 (s, 6H, CH$_3$, Xyl-B), 6.23 (t, 1H, H$^4$, Xyl-C, J = 7.5), 6.33 (t, 1H, H$^4$, Xyl-B, J = 7.6), 6.78 (d, 2H, H$^{3,5}$, Xyl-C, J = 7.5), 6.83 (d, 2H, H$^{3,5}$, Xyl-B, J = 7.6), 6.93–6.96 (m, 3H, H$^{3,5}$, Xyl-D and 1H, H$^5$, thiazole), 7.03 (d, 2H, H$^{3,5}$, Xyl-A, J = 7.6), 7.11 (t, 1H, H$^4$, Xyl-D, J = 7.6), 7.19 (t, 1H, H$^4$, Xyl-A, J = 7.6), 8.07 (d, 1H, H$^4$, thiazole, J = 4.0). $^{13}$C{$^1$H} NMR ($\delta$, ppm): 18.43 (2C, CH$_3$, Xyl-D), 18.50 (2C, CH$_3$, Xyl-A), 19.09 (2C, CH$_3$, Xyl-B), 19.42 (2C, CH$_3$, Xyl-C), 113.56 (1C, C$^5$, thiazole), 124.01 (1C, C$^4$, Xyl-C), 127.34 (2C, C$^{3,5}$, Xyl-D), 127.38 (2C, C$^{3,5}$, Xyl-B), 127.57 (1C, C$^4$, Xyl-B), 127.61 (2C, C$^{3,5}$, Xyl-A), 127.69 (2C, C$^{3,5}$, Xyl-C), 127.97 (2C, C$^{2,6}$, Xyl-C), 128.95 (1C, C$^4$, Xyl-D), 129.48 (1C, C$^4$, Xyl-A), 132.63 (2C, C$^{2,6}$, Xyl-B), 134.13 (2C, C$^{2,6}$, Xyl-A), 134.18 (1C, C$^4$, thiazole), 134.36 (2C, C$^{2,6}$, Xyl-D), 148.16 (1C, C$^1$, Xyl-B), 148.40 (1C, C$^1$, Xyl-C), 160.23 (1C, C$^2$, carbene), 161.76 (1C, C$^2$, thiazole), 179.22 (1C, C$^1$, carbene). The signals of the isocyanide quaternary atoms and C$^1$ Xyl-A and Xyl-D were not found even at high acquisition.

**4a/b mixtures**. Anal. calcd for C$_{39}$H$_{38}$Br$_2$N$_6$Pd$_2$S: C, 47.05; H, 3.85; N, 8.44, found: C, 46.70; H, 3.76; N, 8.92. HRESI$^+$ MS: calcd for C$_{39}$H$_{38}$BrN$_6$Pd$_2$S$^+$ 915.0124, found m/z 915.0136 [M − Br]$^+$.

**4a**. Yield 90% (27 mg). IR (KBr, selected bands, cm$^{-1}$): $\nu$(C–H) 2921 (w), 2920 (w), 2853 (w), $\nu$(C≡N) 2206 (s), $\nu$(C=N) 1648 (s), $\delta$(C–H) 771 (m), 638 (s). $^1$H NMR ($\delta$, ppm, J/Hz): 2.02 (s, 6H, CH$_3$, Xyl-D), 2.24 (s, 6H, CH$_3$, Xyl-C), 2.29 (s, 6H, CH$_3$, Xyl-A), 2.45 (s, 6H, CH$_3$, Xyl-B), 6.18 (t, 1H, H$^4$, Xyl-C, J = 7.6), 6.48 (t, 1H, H$^4$, Xyl-B, J = 7.6), 6.66 (d, 2H, H$^{3,5}$, Xyl-C, J = 7.6), 6.90 (d, 2H, H$^{3,5}$, Xyl-B, J = 7.6), 6.96 (d, 2H, H$^{3,5}$, Xyl-D, J = 7.6), 7.01 (d, 1H, H$^5$, thiazole, J = 4.0), 7.06 (d, 2H, H$^{3,5}$, Xyl-A, J = 7.6), 7.13 (t, 1H, H$^4$, Xyl-D, J = 7.6), 7.23 (m, 1H, H$^4$, Xyl-A, J = 7.6), 7.47 (s, 1H, H$^4$, thiazole, J = 4.0). $^{13}$C{$^1$H} NMR ($\delta$, ppm): 18.13 (2C, CH$_3$, Xyl-A), 18.21 (2C, CH$_3$, Xyl-D), 19.10 (2C, CH$_3$, Xyl-C), 19.37 (2C, CH$_3$, Xyl-B), 114.59 (1C, C$^5$, thiazole), 118.07 (1C, C$^1$, Xyl-D), 120.61 (1C, C$^1$, Xyl-A), 124.43 (1C, C$^4$, Xyl-C), 126.24 (2C, C$^{2,6}$, Xyl-C), 127.48 (2C, C$^{3,5}$, Xyl-D), 127.79 (2C, C$^{3,5}$, Xyl-C), 127.90 (2C, C$^{3,5}$, Xyl-A), 128.12 (2C, C$^{3,5}$, Xyl-B), 129.64 (1C, C$^4$, Xyl-D), 130.36 (1C, C$^4$, Xyl-A), 130.40 (1C, C$^4$, Xyl-B), 134.38 (1C, C$^4$, thiazole), 134.73 (2C, C$^{2,6}$, Xyl-A), 134.83 (2C, C$^{2,6}$, Xyl-D), 136.16 (2C, C$^{2,6}$, Xyl-B), 141.64 (1C, C$^1$, Xyl-B), 147.91 (1C, C$^1$, Xyl-C), 150.89 (1C, C$^2$, carbene), 176.70 (1C, C$^2$, thiazole), 185.68 (1C, C$^1$, carbene). The signals of the isocyanide quaternary atoms were not found even at high acquisition.

**4b**. Yield 86% (26 mg). IR (KBr, selected bands, cm$^{-1}$): $\nu$(C–H) 2977 (w), 2920 (w), 2851 (w), $\nu$(C≡N) 2202 (s), $\nu$(C=N) 1627 (s), 1556 (s), $\delta$(C–H) 773 (m), 747 (m). $^1$H NMR ($\delta$, ppm, J/Hz): 2.25 (s, 12H, CH$_3$, Xyl-C and Xyl-D), 2.28 (s, 6H, CH$_3$, Xyl-A), 2.44 (s, 6H, CH$_3$, Xyl-B), 6.31 (t, 1H, H$^4$, Xyl-C, J = 7.7), 6.40 (t, 1H, H$^4$, Xyl-B, J = 7.6), 6.80–6.84 (m, 4H, H$^{3,5}$,

Xyl-B and $H^{3,5}$, Xyl-C), 6.95 (d, 2H, $H^{3,5}$, Xyl-D, J = 7.8), 7.04–7.06 (m, 3H, 1H, $H^5$, thiazole, and $H^{3,5}$, Xyl-A), 7.13 (t, 1H, $H^4$, Xyl-D, J = 7.8), 7.22 (t, 1H, $H^4$, Xyl-A, J = 7.7), 7.85 (s, 1H, $H^4$, thiazole). $^{13}$C{$^1$H} NMR (δ, ppm): 18.14 (2C, $CH_3$, Xyl-A), 18.25 (2C, $CH_3$, Xyl-D), 19.39 (2C, $CH_3$, Xyl-C), 19.54 (2C, $CH_3$, Xyl-B), 115.27 (1C, $C^5$, thiazole), 117.63 (1C, $C^1$, Xyl-A), 120.80 (1C, $C^1$, Xyl-D), 124.97 (1C, $C^4$, Xyl-C), 126.29 (1C, CNXyl-D), 127.50 (2C, $C^{3,5}$, Xyl-D), 127.70 (2C, $C^{2,6}$, Xyl-C), 127.84 (2C, $C^{3,5}$, Xyl-A), 127.97 (2C, $C^{3,5}$, Xyl-B), 128.03 (2C, $C^{3,5}$, Xyl-C), 128.35 (1C, $C^4$, Xyl-B), 129.80 (1C, $C^4$, Xyl-D), 130.35 (1C, $C^4$, Xyl-A), 132.59 (2C, $C^{2,6}$, Xyl-B), 133.93 (1C, $C^4$, thiazole), 134.77 (2C, $C^{2,6}$, Xyl-D), 135.00 (2C, $C^{2,6}$, Xyl-A), 145.41 (1C, $C^1$, Xyl-B), 147.10 (1C, $C^1$, Xyl-C), 148.25 (1C, $C^2$, carbene), 161.29 (1C, $C^2$, thiazole), 168.88 (1C, $C^1$, carbene). The signal of the isocyanide quaternary atom CNXyl-A was not found even at high acquisition.

**5a/b mixtures**. Anal. calcd for $C_{39}H_{38}I_2N_6Pd_2S \cdot 0.1CHCl_3$: C, 42.64; H, 3.49; N, 7.63, found: C, 43.03; H, 3.42; N, 7.12. HRESI$^+$ MS: calcd for $C_{39}H_{38}IN_6Pd_2S^+$ 962.9988, found $m/z$ 962.9960 [M – I]$^+$.

**5a**. Yield 92% (30 mg). IR (KBr, selected bands, cm$^{-1}$): ν(C–H) 2922 (w), 2852 (w), 2851 (w), ν(C≡N) 2203 (s), ν(C=N) 1650 (m), δ(C–H) 773 (m), 638 (s). $^1$H NMR (δ, ppm, J/Hz): 2.02 (s, 6H, $CH_3$, Xyl-D), 2.25 (s, 6H, $CH_3$, Xyl-C), 2.29 (s, 6H, $CH_3$, Xyl-A), 2.45 (s, 6H, $CH_3$, Xyl-B), 6.18 (t, 1H, $H^4$, Xyl-C, J = 7.6), 6.48 (t, 1H, $H^4$, Xyl-B, J = 7.6), 6.66 (d, 2H, $H^{3,5}$, Xyl-C, J = 7.6), 6.91 (d, 2H, $H^{3,5}$, Xyl-B, J = 7.6), 6.96 (d, 2H, $H^{3,5}$, Xyl-D, J = 7.6), 7.01 (d, 1H, $H^5$, thiazole, J = 4.0), 7.06 (d, 2H, $H^{3,5}$, Xyl-A, J = 7.6), 7.13 (t, 1H, $H^4$, Xyl-D, J = 7.6), 7.23 (t, 1H, $H^4$, Xyl-A, J = 7.6), 7.47 (d, 1H, $H^4$, thiazole, J = 4.0). $^{13}$C{$^1$H} NMR (δ, ppm): 18.12 (2C, $CH_3$, Xyl-D), 18.21 (2C, $CH_3$, Xyl-A), 19.09 (2C, $CH_3$, Xyl-B), 19.37 (2C, $CH_3$, Xyl-C), 114.56 (1C, $C^5$, thiazole), 118.07 (1C, $C^1$, Xyl-D), 120.61 (1C, $C^1$, Xyl-A), 124.43 (1C, $C^4$, Xyl-C), 126.23 (2C, $C^{2,6}$, Xyl-C), 127.47 (2C, $C^{3,5}$, Xyl-D), 127.78 (2C, $C^{3,5}$, Xyl-C), 127.89 (2C, $C^{3,5}$, Xyl-A), 128.11 (2C, $C^{3,5}$, Xyl-B), 129.63 (1C, $C^4$, Xyl-D), 130.35 (1C, $C^4$, Xyl-A), 130.38 (1C, $C^4$, Xyl-B), 134.40 (1C, $C^4$, thiazole), 134.71 (2C, $C^{2,6}$, Xyl-A), 134.82 (2C, $C^{2,6}$, Xyl-d), 136.15 (2C, $C^{2,6}$, Xyl-B), 141.65 (1C, $C^1$, Xyl-B), 147.90 (1C, $C^1$, Xyl-C), 150.87 (1C, $C^2$, carbene), 176.72 (1C, $C^2$, thiazole), 185.76 (1C, $C^1$, carbene). The signals of the isocyanide quaternary atoms were not found even at high acquisition.

**5b**. Yield 88% (29 mg). IR (KBr, selected bands, cm$^{-1}$): ν(C–H) 2975 (w), 2920 (w), 2850 (w), ν(C≡N) 2201 (s), ν(C=N) 1629 (s), 1555 (s), ν(C–H) 771 (m). $^1$H NMR (δ, ppm, J/Hz): 2.02 (s, 12H, $CH_3$, Xyl-D and Xyl-C), 2.29 (s, 6H, $CH_3$, Xyl-A), 2.45 (s, 6H, $CH_3$, Xyl-B), 6.28 (t, 1H, $H^4$, Xyl-C, J = 7.6), 6.38 (t, 1H, $H^4$, Xyl-B, J = 7.6), 6.81 (d, 2H, $H^{3,5}$, Xyl-C, J = 7.6), 6.84 (d, 2H, $H^{3,5}$, Xyl-B, J = 7.6), 6.95 (m, 2H, $H^{3,5}$, Xyl-D), 7.04–7.08 (m, 3H, 2H$^{3,5}$, Xyl-A and $H^5$, thiazole), 7.13 (t, 1H, $H^4$, Xyl-D, J = 7.6), 7.22 (t, 1H, $H^4$, Xyl-A, J = 7.7), 7.78 (s, 1H, $H^4$, thiazole). $^{13}$C{$^1$H} NMR (δ, ppm): 18.17 (2C, $CH_3$, Xyl-A), 18.24 (2C, $CH_3$, Xyl-D), 19.38

(2C, CH$_3$, Xyl-C), 19.51 (2C, CH$_3$, Xyl-B), 115.09 (1C, C$^5$, thiazole), 117.62 (1C, C$^1$, Xyl-A), 120.79 (1C, C$^1$, Xyl-D), 124.92 (1C, C$^4$, Xyl-C), 126.16 (1C, CNXyl-D), 127.47 (2C, C$^{3,5}$, Xyl-D), 127.68 (2C, C$^{2,6}$, Xyl-C), 127.79 (2C, C$^{3,5}$, Xyl-A), 127.94 (2C, C$^{3,5}$, Xyl-B), 128.11 (2C, C$^{3,5}$, Xyl-C), 128.55 (1C, C$^4$, Xyl-B), 129.74 (1C, C$^4$, Xyl-D), 130.23 (1C, C$^4$, Xyl-A), 132.48 (2C, C$^{2,6}$, Xyl-B), 133.98 (1C, C$^4$, thiazole), 134.67 (2C, C$^{2,6}$, Xyl-D), 134.98 (2C, C$^{2,6}$, Xyl-A), 145.34 (1C, C$^1$, Xyl-B), 147.14 (1C, C$^1$, Xyl-C), 148.67 (1C, C$^2$, carbene), 161.32 (1C, C$^1$, carbene). The signals of the isocyanide quaternary atom CNXyl-A and C$^2$ thiazole were not found even at high acquisition.

### 3.4. Data for the Intermediate Complexes **6a** and **6b**

**6a**. HRESI$^+$ MS: calcd for C$_{40}$H$_{38}$F$_3$N$_6$O$_3$Pd$_2$S$_2^+$ 985.0468, found *m/z* 985.0444 [M − OTf]$^+$. $^1$H NMR ($\delta$, ppm, J/Hz): 2.02 (s, 6H, CH$_3$, Xyl-D), 2.25 (s, 6H, CH$_3$, Xyl-C), 2.29 (s, 6H, CH$_3$, Xyl-A), 2.45 (s, 6H, CH$_3$, Xyl-B), 6.18 (t, 1H, H$^4$, Xyl-C, J = 7.6), 6.49 (t, 1H, H$^4$, Xyl-B, J = 7.6), 6.66 (d, 2H, H$^{3,5}$, Xyl-C, J = 7.6), 6.90 (d, 2H, H$^{3,5}$, Xyl-B, J = 7.6), 6.96 (m, 2H, H$^{3,5}$, Xyl-D), 7.04–7.08 (d, 1H, H$^5$, thiazole, J = 4.1), 7.04–7.08 (d, 2H, H$^{3,5}$, Xyl-A, J = 7.7), 7.13 (t, 1H, H$^4$, Xyl-D, J = 7.6), 7.24 (m, 1H, H$^4$, Xyl-A), 7.48 (d, 1H, H$^4$, thiazole, J = 4.1).

**6b**. HRESI$^+$ MS: calcd for C$_{40}$H$_{38}$F$_3$N$_6$O$_3$Pd$_2$S$_2^+$ 985.0468, found *m/z* 985.0444 [M − OTf]$^+$. $^1$H NMR ($\delta$, ppm, J/Hz): 2.25 (s, 12H, CH$_3$, Xyl-D and Xyl-C), 2.29 (s, 6H, CH$_3$, Xyl-A), 2.45 (s, 6H, CH$_3$, Xyl-B), 6.29 (t, 1H, H$^4$, Xyl-C, J = 7.6), 6.49 (t, 1H, H$^4$, Xyl-B, J = 7.6), 6.81 (d, 2H, H$^{3,5}$, Xyl-C, J = 7.6), 6.83 (d, 2H, H$^{3,5}$, Xyl-B, J = 7.6), 6.96 (d, 2H, H$^{3,5}$, Xyl-D, J = 7.6), 7.05 (d, 2H, H$^{3,5}$, Xyl-A, J = 7.7), 7.09 (d, 1H, H$^5$, thiazole, J = 4.1), 7.14 (t, 1H, H$^4$, Xyl-D, J = 7.6), 7.23 (t, 1H, H$^4$, Xyl-A J = 7.7), 7.72 (d, 1H, H$^4$, thiazole, J = 4.1).

### 3.5. X-ray Diffraction

Single crystal X-ray diffraction experiments for all crystals were carried out using an Oxford Diffraction "Xcalibur" diffractometer with monochromated MoK$\alpha$ radiation. The crystals were kept at 100 K throughout data collection. Initial data processing was carried out using the CrysAlisPro (Agilent Technologies, ver. 1.171.36.32) data reduction program. The structures were solved using the Superflip [57] structure solution program and refined by means using the ShelXL [58] structure refinement program incorporated in the Olex2 [59] program package. All crystallographic data for this paper can be obtained free of charge via the Cambridge Crystallographic Database (CCDC 2263951, 2263952).

### 3.6. Computational Details

The quantum chemical computations for model compounds were performed utilizing the DFT-functional M06 [60] (it was specifically developed to describe weak dispersion forces and non-covalent interactions) using Gaussian-09 [61] software. The experimentally

obtained XRD geometries were used as initial points. The quantum chemical computations were carried out using the quasi-relativistic Stuttgart effective core potentials that considered 28 or 46 core electrons and the corresponding basis sets [62] for the Pd or I atoms, respectively, and the 6-31 + G* basis sets for other atoms. The continuum solvation model SMD [63], with chloroform as the solvent, was used to take into account the solvent effects. The Hessian matrices were analytically calculated for the optimized model structures (no symmetry restrictions were applied in all cases) in order to prove the location of true minima on the potential energy surface (no imaginary frequencies were found in all cases), and to estimate the thermodynamic parameters at 25 °C and 1 atm. The AIM analysis [53] was carried out using Multiwfn 3.3.4 software [64]. The atomic coordinates for optimized equilibrium model structures are presented in the Supplementary Materials, Table S2.

## 4. Conclusions

Binuclear diaminocarbene complexes, which form as the regioisomer mixture **3a/b** in the reaction of *cis*-[PdCl$_2$(CNXyl)$_2$] with 1,3-thiazol-2-amine, are able to exchange an anionic chloride ligand with other halides, such as Br or I. The exchange occurs upon the sequential addition of first AgOTf and then KBr or KI to a mixture of complex **3**. This process also affords binuclear complexes **4a/b** or **5a/b** as mixtures of kinetically (**a**) and thermodynamically (**b**) controlled regioisomers. All these species were characterized by elemental analysis, MS, and NMR in CDCl$_3$ solutions. The solid-state structures of two complexes—the thermodynamically controlled regioisomers **4b** and **5b**—were elucidated with X-ray diffraction.

In CDCl$_3$ solutions, we observed interconversion of kinetically (**a**) and thermodynamically (**b**) controlled regioisomers affording equilibrium mixtures of the isomers. Compared to the previously described chloride complex **3**, this equilibrium for the bromide complex **4** and for the iodide complex **5** is, firstly, established much faster, and secondly, shifted to a much greater extent towards the thermodynamically controlled isomer. The results of the quantum chemical computations together with AIM analysis showed that in a chloroform solution, each pair of the isomers exhibited two different types of chalcogen bonding, such as S···X (**4a** and **5a**) and S···N (**4b** and **5b**); the presence of CBs for **4b** and **5b** in the solid state was experimentally proven by XRD analysis as well.

Based on the combined experimental and theoretical data, it can be concluded that thermodynamic favorability for the formation of thermodynamically controlled regioisomers increases in the row Cl<Br≈I and correlates well with the energy difference between S···N and S···X (where X = Cl, Br, or I) chalcogen bonds in kinetically and thermodynamically controlled products. This means that it is possible to change the structure of metallocycles in binuclear diaminocarbene complexes by simply replacing one halide ligand with another.

**Supplementary Materials:** The following supporting information can be downloaded at: https: //www.mdpi.com/article/10.3390/inorganics11060255/s1, Figures S1–S8: NMR $^1$H, and $^{13}$C spectra of complexes **4**–**5a,b**; Tables S1 and S2: Crystal data and structure refinement for complexes **4b**, **5b** and cartesian atomic coordinates of optimized equilibrium model structures; Crystallographic information files (CIF) and checkCIF report files for complexes **4b**, **5b**.

**Author Contributions:** Conceptualization, V.P.B.; methodology, V.P.B.; investigation, R.A.P., A.S.N. and V.V.S.; writing—original draft preparation, A.S.N. and R.A.P.; writing—review and editing, V.P.B.; visualization, V.V.S.; supervision, V.P.B. All authors have read and agreed to the published version of the manuscript.

**Funding:** This research was funded by the Russian Science Foundation, grant number 19-13-00008.

**Data Availability Statement:** The data presented in this study are available on request from the corresponding author.

**Acknowledgments:** Physicochemical studies were performed at the Center for Magnetic Resonance, the Center for X-ray Diffraction Studies, and the Center for Chemical Analysis and Materials Research of Saint Petersburg State University.

**Conflicts of Interest:** The authors declare no conflict of interest. The funders had no role in the design of the study; in the collection, analyses, or interpretation of data; in the writing of the manuscript; or in the decision to publish the results.

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
