# Peer review of "Molecular Switching through Chalcogen-Bond-Induced Isomerization of Binuclear (Diaminocarbene)PdII Complexes"

_inorganics, doi:10.3390/inorganics11060255_

Round 1

Reviewer 1 Report

Manuscript: Molecular Switching through Chalcogen Bond-Induced 2 Isomerization of Binuclear (Diaminocarbene)PdII Complexes

Authors: Roman A. Popov, Alexander S. Novikov, Vitalii V. Suslonov and Vadim P. Boyarskiy.

Authors describe interesting process of isomerization with different Pd coordination modes. This process was described earlier in JACS and Inorg. Chim. Acta, now the process is extended on change of halide ligand (Cl into Br and I) on Pd atoms. Structures were solved correctly. The presented data are in good relation to the manuscript concept. The manuscript is interesting, but at the moment Authors done something very strange with the elemental analysis. This is why the manuscript require second check, thus I suggest major revision, as indicated below and after revision second check if manuscript can be accepted finally.

1.     Could Authors give some comments why spectrum of the bromo isomer (4) gave so different 1H NMR spectrum (compared to Cl and I isomers)? Why singlets and not doublets above 7.4 ppm are observed?

2.     Please check and correct values for C, N and H calculated for 3a and 3b, as now they are wrong. The same problem is with the analysis for 5a and 5b - are wrongly calculated (for example C calc. 42.06 and not 42.6; N 7.55 and not 7.63). For such formula found values are above experimental error for 5a and 5 b (C 43.03, almost 1% of error).

3.     Please explain why analysis of isomers (found), are always identical? Authors should indicate for which isomer they done measurements and for other isomer that elemental analysis were not performed. Or that analysis were for mixture of isomers (compound 3, 4 and 5) and that analysis for 3a, 3b, 4a, 4b, 5a and 5b were not done. This must be changed.

Author Response

1) Remark: Could Authors give some comments why spectrum of the bromo isomer (4) gave so different 1H NMR spectrum (compared to Cl and I isomers)? Why singlets and not doublets above 7.4 ppm are observed?

Our answer: It is not surprising that the structure of the thiazole proton signal in the chloride complex differs from its structure in the bromide complex. Its chemical shift is also very different. And this is obviously due to a significant difference in the distribution of electron density in different complexes. It is stranger that the structure of the signal differs in the bromide and iodide complexes, although both the chemical shifts of this proton and the relative stabilities of regioisomers b (which is an indirect evidence of the nature of the electron density distribution in the complexes) are similar. We were also intrigued by this, and we tried to determine whether, in the case of the bromide complex, there is some kind of an exchange process that leads to signal broadening. However, temperature experiments did not show the existence of such a process. When the temperature was lowered down to –20 °C, we did not observe the splitting of the broadened singlet into components. It is possible that the different structures of the thiazole proton signals are due to the fact that the electronic structures of the bromide and iodide complexes are still different.

2) Remark: Please check and correct values for C, N and H calculated for 3a and 3b, as now they are wrong. The same problem is with the analysis for 5a and 5b - are wrongly calculated (for example C calc. 42.06 and not 42.6; N 7.55 and not 7.63). For such formula found values are above experimental error for 5a and 5 b (C 43.03, almost 1% of error).

Our answer: We once again checked the elemental analysis data for the regioisomer mixture 3a/b. We did not find any errors; the data correspond to the proposed molecular formula, taking into account the residual amount of the solvent in the sample, within the permissible errors: Anal. calcd. for C39H38Cl2N6Pd2S·0.2CDCl3: C, 50.59; H, 4.16; N, 9.03. Found: C, 50.19; H, 4.12; N, 9.43.

In the case of regioisomer mixture 5a/b, we made a typo (the formula should be C39H38I2N6Pd2S·0.1CHCl3 instead of C39H38I2N6Pd2S·0.2CHCl3). Anal. calcd. for C39H38I2N6Pd2S·0.1CHCl3: C, 42.64; H, 3.49; N, 7.63. Found: C, 43.03; H, 3.42; N, 7.12. That is, the result practically fits into the permissible error interval. The result for nitrogen differs not by 0.5%, but by 0.51%, but we hope that this will not become an obstacle to publication.

3) Remark: Please explain why analysis of isomers (found), are always identical? Authors should indicate for which isomer they done measurements and for other isomer that elemental analysis were not performed. Or that analysis were for mixture of isomers (compound 3, 4 and 5) and that analysis for 3a, 3b, 4a, 4b, 5a and 5b were not done. This must be changed.

Our answer: The reviewer is right; we have not accurately described the characterization procedure. Of course, elemental analysis and recording of mass spectra were performed for the mixtures of regioisomers. We apologize for the inaccurate description. Necessary clarifications are included in the text of the experimental part.

Reviewer 2 Report

This is a very nice paper. Using a combined experimental and computational approach, the authors investigated the molecular switching through chalcogen bond-induced isomerization of binuclear (diaminocarbene)PdII complexes. The results presented in this study indicated that it is possible to change the structure of metallocycles in binuclear diaminocarbene complexes by simply replacing one halide ligand with another. The study is very comprehensive, and the paper is well written. Publication in Inorganics can be recommended after attention to the issues outlined below:

1) The first paragraph of the introduction section (lines 25-31 on page 1) should be rewritten. I do not understand what the authors want to express to the reader in this paragraph, especially in the last sentences. The abbreviations EWG and Z should be defined the first time they are used. On the other hand, the important paper on the chalcogen bond (J. Phys. Chem. A 2009, 113, 8132–8135) should be cited in this paragraph because it coined the term “chalcogen bond” and defined the chalcogen bond for the first time.

2) In Table 3 on page 3, the superscripts a, b and c should be defined.

3) In lines 202-204 on page 8, the authors stated that “However, one can notice the shortening of S···X contact in the row Cl>Br>I causing rise of energy difference between two types of chalcogen bonds ΔEint (Table 3)”. This sentence should be rewritten. Could the shortening of S···X contact in the row Cl>Br>I cause the increase of interaction energy difference between two types of chalcogen bonds in Table 3? Evidently, the answer is No. It is natural that the S···X contacts shorten in the order of Cl>Br>I because the atomic radii increase in the order of Cl>Br>I.

4) Line 397, page 13: “Gaussian-09” should be “Gaussian 09”.

5) Line 400, page 13: “6-31+G31G*” should be “6-31+G*”.

6) The sentence in lines 410 and 411 on page 14 should be rewritten.

Minor editing of English language required.

Author Response

1) Remark: The first paragraph of the introduction section (lines 25-31 on page 1) should be rewritten. I do not understand what the authors want to express to the reader in this paragraph, especially in the last sentences. The abbreviations EWG and Z should be defined the first time they are used. On the other hand, the important paper on the chalcogen bond (J. Phys. Chem. A 2009, 113, 8132–8135) should be cited in this paragraph because it coined the term “chalcogen bond” and defined the chalcogen bond for the first time.

Our answer: In this paragraph, we briefly introduce the concepts of "σ-hole interactions" and “σ- lone pair interactions”. After all, those chalcogen bonds that we describe further in our manuscript are precisely σ-lone pair interactions (σ-lp). Therefore, we considered this paragraph a necessary preface to the article. We have deciphered the abbreviation EWG. We abandoned the abbreviation Z, replacing it with the word "element". We added the necessary reference in accordance with the Reviewer's request.

2) Remark: In Table 3 on page 3, the superscripts a, b and c should be defined.

Our answer: We apologize for the typo in the table number. Probably the Reviewer had in mind Table 4, since the superscripts are defined in the bottom of Table 3:

aEint = –V(r)/2 [50]; bEint 0.429G(r) [51]; cData taken from Ref. [36]”

We have simplified Table 4 in the new version of the manuscript, there is now only one superscripts a, it is defined.

3) Remark: In lines 202-204 on page 8, the authors stated that “However, one can notice the shortening of S···X contact in the row Cl>Br>I causing rise of energy difference between two types of chalcogen bonds ΔEint (Table 3)”. This sentence should be rewritten. Could the shortening of S···X contact in the row Cl>Br>I cause the increase of interaction energy difference between two types of chalcogen bonds in Table 3? Evidently, the answer is No. It is natural that the S···X contacts shorten in the order of Cl>Br>I because the atomic radii increase in the order of Cl>Br>I.

Our answer: In accordance with the Reviewer’s comment, the appropriate sentence was corrected and rewritten.

4) Remark: Line 397, page 13: “Gaussian-09” should be “Gaussian 09”.

Our answer: Done, thank you!

5) Remark: Line 400, page 13: “6-31+G31G*” should be “6-31+G*”.

Our answer: Done, thank you!

6) Remark: The sentence in lines 410 and 411 on page 14 should be rewritten.

Our answer: We have rewritten the corresponding sentence.

Round 2

Reviewer 1 Report

Authors changed the complex composition (from 0.2 CHCl3 into 0.1 CHCl3) and now the analysis fits better to the complex composition. Authors changed also the description, that the analysis were done for only mixture of isomers and not for pure isomers separatelly.

All other problems were removed and manuscript can be accepted in present form.